# Phenotypic Antimicrobial Susceptibility of *Escherichia coli* from Raw Meats, Ready-to-Eat Meats, and Their Related Samples in One Health Context

**DOI:** 10.3390/microorganisms9020326

**Published:** 2021-02-05

**Authors:** Frederick Adzitey, Nurul Huda, Amir Husni Mohd Shariff

**Affiliations:** 1Department of Food Science and Technology, University for Development Studies, P. O. Box TL 1882 Tamale, Ghana; 2Faculty of Food Science and Nutrition, Universiti Malaysia Sabah, Jalan UMS, Kota Kinabalu 88400, Malaysia; amir.husni@ums.edu.my

**Keywords:** antimicrobial resistance, bacteria, food chain, humans, meat, one-health

## Abstract

Meat is an important food source that can provide a significant amount of protein for human development. The occurrence of bacteria that are resistant to antimicrobials in meat poses a public health risk. This study evaluated the occurrence and antimicrobial resistance of *E. coli* (*Escherichia coli*) isolated from raw meats, ready-to-eat (RTE) meats and their related samples in Ghana. *E. coli* was isolated using the USA-FDA Bacteriological Analytical Manual and phenotypic antimicrobial susceptibility test was performed by the disk diffusion method. Of the 200 examined meats and their related samples, 38% were positive for *E. coli*. Notably, *E. coli* was highest in raw beef (80%) and lowest in RTE pork (0%). The 45 *E. coli* isolates were resistant ≥ 50% to amoxicillin, trimethoprim and tetracycline. They were susceptible to azithromycin (87.1%), chloramphenicol (81.3%), imipenem (74.8%), gentamicin (72.0%) and ciprofloxacin (69.5%). A relatively high intermediate resistance of 33.0% was observed for ceftriaxone. *E. coli* from raw meats, RTE meats, hands of meat sellers and working tools showed some differences and similarities in their phenotypic antimicrobial resistance patterns. Half (51.1%) of the *E. coli* isolates exhibited multidrug resistance. The *E. coli* isolates showed twenty-two different resistant patterns, with a multiple antibiotic resistance index of 0.0 to 0.7. The resistant pattern amoxicillin (A, *n* = 6 isolates) and amoxicillin-trimethoprim (A-TM, *n* = 6 isolates) were the most common. This study documents that raw meats, RTE meats and their related samples in Ghana are potential sources of antimicrobial-resistant *E. coli* and pose a risk for the transfer of resistant bacteria to the food chain, environment and humans.

## 1. Introduction

*Escherichia coli* is a Gram-negative bacteria of the enterobacteriaceae family [1]. *Escherichia coli* lives in the gastrointestinal tracts (GIT) of humans and animals [1,2]. They commonly end up in the environment due to the close association of animals and humans to their environment [1,3]. The interaction between animals and humans also paves the way for transfer of these bacteria between them [4]. Although most *E. coli* strains are noted to be commensals, pathogenic *E. coli* exists [2]. Pathogenic *E. coli* strains have been responsible for the cause of food poisoning, pneumonia and urinary tract infections [1]. *E. coli* infections can also lead to hospitalizations, loss of manpower and even death [5,6]. For instance, in the USA, the Center for Control and Disease Prevention (CDC) [6] reported an *E. coli* infection from an unknown food that caused 16 people to be ill, with eight hospitalizations resulting in one developing hemolytic uremic syndrome and one death. Outbreaks of *E. coli* infections in Europe have also been noted [7]. The contribution of some strains of *E. coli* to travelers’ diarrhea, especially among children in lower income countries, including Ghana, cannot be overemphasized, although official reports are limited [1,8].

*Escherichia coli* infections in humans normally result from the consumption of contaminated foods that are present in the food chain. Foods of animal origin (meat and meat products), in particular, have played a major role in the spread of *E. coli* infections in humans. For example, beef, chevon, pork, poultry and/or their products have been implicated in the cause of foodborne infections in the USA and Europe [5,6,9]. These meat and meat products, to a lesser extent, also serve as sources for which *E. coli* can cross contaminate other foods and the environment. The European Food Safety Authority (EFSA) [9] reported that meat and meat products were the food category that were tested the most for *E. coli* linked to human infections in a five-year period in Europe. The occurrence of *E. coli* in raw meats, ready-to-eat (RTE) meats, humans, working tools and/or the environment have been reported in America [5,6,10], Europe [9,11], Asia [12,13,14] and Africa [15,16,17]. *Escherichia coli* infections are usually self-limiting; however, when treatments are required, antimicrobials are used [18]. 

Antimicrobials are agents used to curtail the growth of microorganisms or are used to kill them [19,20]. Antimicrobial resistance occurs when bacteria resist antimicrobials that are meant to destroy them. The unabated use of antimicrobials in farm animals for growth promotion, treatment of sick animals or as prophylactics has been the major cause of the spread of antimicrobial resistances in animals. This can spread to humans via consumption of meat and meat products from farm animals containing resistance bacteria. Humans and animals can cross contaminate other foods and the environment with resistant bacteria when they come into contact with them. There is also the possibility of resistant bacteria to spread across the food chain, humans and animals from the environment. The resistance of bacteria, including *E. coli*, to antimicrobials has received much attention in recent times due to the emergence of multidrug resistance and the difficulty in destroying such bacteria when they are involved in infections [18,19]. The use of antimicrobials in animal production in Ghana is not strictly regulated and monitored. As such farmers and farm attenders sometimes treat their animals with antimicrobials without prescription or adherence to instructed dosage and withdrawal periods [21,22]. In addition, farmers did not demonstrate in-depth knowledge of the antimicrobials used [21,22]. These practices pose the risk of bacteria developing resistance to antimicrobials. Furthermore, studies have shown the presence of antimicrobial resistance *E. coli* in meats and meats products in Ghana [23,24,25]. Dsani et al. [23] reported that *E. coli* isolated from meat sources were resistant to cefuroxime (17%), sulfamethoxazole-trimethoprim (21%), tetracycline (45%) and ampicillin (57%). Eibach et al. [24] found that *E. coli* isolates from poultry meat were resistant to ciprofloxacin (21%) and sulfamethoxazole-trimethoprim (32%), and harbored resistance genes such as Bla_SHV_, bla_CTX-M_ and bla_TEM_. Resistance of *E. coli* isolated from beef to tetracycline, azithromycin and sulfamethoxazole-trimethoprim was 40%, 33% and 20%, respectively [24]. They also reported that some of the *E. coli* isolates were resistant to multiple antimicrobials and/or harbored Bla_SHV_, bla_CTX-M,_ bla_TEM,_
*tet (A)*, *aph (3’’)-Ib*, *apha (6), dfrA14* and *mdf (A)* resistance genes [23,24,25]. Thus, meats and meat products in Ghana are potential sources of antimicrobial-resistant bacteria.

Moreover, studies on the spread of *E. coli* among meats, humans and their environment in Ghana are limited. Studies have also shown that the manner in which animals are handled prior to slaughter, during slaughter and post-slaughter makes meat and their related samples potential sources of foodborne pathogens [26,27,28,29,30,31,32]. In addition, the sale of meats and meat products is sometimes conducted in open markets and by roadsides without adherence to strict hygiene [33,34,35]. RTE meats in particular are consumed without prior heating [36]. With all these handling activities, meat and meat products expose consumers to foodborne infections and antimicrobial resistance bacteria. Therefore, this study is aimed at determining whether raw meats, RTE meats and their related samples (hands of meat sellers, knives for cutting meat, tables on which meats are placed for cutting and for sale, utensils used by meat sellers) are contaminated by antimicrobial-resistant *E. coli*.

## 2. Materials and Methods

### 2.1. Study Area

This study was carried out in the Upper East Region of Ghana. The region shares a boundary with Burkina Faso to the north, the Republic of Togo to the east, West Mamprusi in the Northern Region to the south and Sissala in the Upper West Region to the west. The region covers an area of 8842 square kilometers and is also located between longitude 0° and 1° West, and latitudes 10°30′ North and 11° North [37].

### 2.2. Duration and Sampling of Raw Meats, Ready-to-Eat (RTE) Meats and Their Related Samples

Sampling was conducted between June and November, 2020. In all, 200 samples were randomly selected and examined for the presence of *E. coli*. The samples consisted of 10 each of raw beef, raw chevon, raw chicken, raw guinea fowl, raw mutton, raw pork, RTE beef, RTE chevon, RTE chicken, RTE guinea fowl, RTE mutton, RTE pork, hand swabs of raw meat sellers, knife swabs from raw meat sellers, table swabs from raw meat sellers, utensil swabs from raw meat sellers, hand swabs from RTE meat sellers, knife swabs from RTE meat sellers, table swabs from RTE meat sellers and utensil swabs from RTE meat sellers. For the raw and RTE meats, 10 g samples were collected into sterile bags, while for hand, knife, table and utensil samples, approximately 10 cm^2^ was swabbed. Raw meats, RTE meats and their related samples were transported in an ice chest containing ice and analyzed immediately on arrival at the laboratory. 

### 2.3. Detection and Confirmation of Escherichia coli in Raw Meats, RTE Meats and Their Related Samples

The isolation of *E. coli* was carried out according to Feng et al. [2]. In total, 10 g of raw meats or RTE meats was pre-enriched in 90 mL of peptone buffered water (MAST Limited, Liverpool, UK) and incubated aerobically at 37 °C for 24 h. After which, the aliquots were streaked on Levine’s eosin-methylene blue (LEMB) agar (MAST Limited, Liverpool, UK) and incubated aerobically at 37 °C for 24 h. Presumptive *E. coli* isolates were purified on trypticase soy agar (Oxoid Limited, Basingstoke, UK) and confirmed using Gram stain (Oxoid Limited, Basingstoke, UK), growth in brilliant green bile (Oxoid Limited, Basingstoke, UK) containing Durham tubes, and *E. coli* latex agglutination test (Oxoid Limited, Basingstoke, UK) [2,30,36]. Prevalence results from *E. coli* were analyzed using binary logistic of IBM Statistical Package for the Social Sciences (SPSS) Version 20, and where significant differences existed, they were separated using Wald chi-square at 5% significance level.

### 2.4. Phenotypic Antimicrobial Resistance Test

The phenotypic antimicrobial susceptibility test was conducted according to the disk diffusion method [38]. In total, 45 *E. coli* isolates were subjected to this test using the following antimicrobials: amoxicillin 30 μg (A), azithromycin 15 µg (ATH), ceftriaxone 30 µg (CRO), chloramphenicol 30 µg (C), ciprofloxacin 5 µg (CIP), gentamicin 10 µg (GM), trimethoprim 2.5 µg (TM), tetracycline 30 ug (T) and imipenem 10 µg (IMI) purchased from MAST Limited, Liverpool, UK. Pure cultures of *E. coli* were grown in 10 mL trypticase soy broth (Oxoid Limited, Basingstoke, UK) overnight at 37 °C and the concentration adjusted to 0.5 McFarland solution. The adjusted solution was spread on Muller Hinton agar (Oxoid Limited, Basingstoke, UK), antimicrobial discs were placed on it and incubated overnight at 37 °C. After incubation, the inhibition zones were measured, and the results interpreted using the Clinical and Laboratory Standards Institute Guidelines [39]. The procedure of Krumperman [40] was used to determine the multiple antibiotic resistance index (MAR index) of the *E. coli* isolates. MAR index was defined as a/b where “a” is the number of antibiotics to which a particular isolate was resistant, and “b” is the total number of antibiotics examined. Multidrug resistance was defined as resistant to ≥3 different classes of antimicrobials [41].

## 3. Results

### 3.1. Distribution of Escherichia coli in Raw Meats, Ready-to-Eat (RTE) Meats and Their Related Samples

The distribution of the *E. coli* isolates in the raw meats, RTE meats and their related samples is shown in Table 1. Raw beef (80%) was the most contaminated source, followed by raw mutton (70%), and raw chevon (60%), raw pork (60%) and knife of raw meat sellers (60%). In addition, the prevalence of *E. coli* in raw beef was significantly higher (*p* < 0.05) than the rest of the samples examined except for raw mutton, raw chevon, raw pork and knife of raw meat sellers. Hands of raw meat sellers and their utensils had the least contamination rate of 10% each. *E. coli* was not detected in RTE pork.

### 3.2. Phenotypic Antimicrobial Susceptibility of Escherichia coli Isolated from Raw Meats, RTE Meats and Their Related Samples

The overall phenotypic antimicrobial resistance of the meats and their related samples is presented in Figure 1. The *E. coli* isolates were highly resistant to amoxicillin (70.9%), tetracycline (57.0%) and trimethoprim (55.0%). Considerable rate of intermediate resistance occurred for ceftriaxone (33.3%), amoxicillin (29.1%), ciprofloxacin (21.1%) and gentamicin (19.6%). The *E. coli* isolates were, however, susceptible to azithromycin (87.1%), chloramphenicol (81.3%), imipenem (74.8%), gentamicin (72.0%) and ciprofloxacin (69.5%).

The phenotypic antimicrobial resistance of *E. coli* isolated from raw meats, RTE meats, hands and working tools of meat sellers can be found in Table 2. *Escherichia coli* isolates from these sources were all resistant ≥55% to amoxicillin. *E. coli* isolates isolated from the hands of meat sellers were resistant to tetracycline (60.0%) and trimethoprim (60.0%), while those isolated from working tools showed 76.9% resistance each to tetracycline and trimethoprim. Resistance to trimethoprim and tetracycline were 52.9% and 41.2%, respectively, for *E. coli* isolates obtained from raw meats. It was 30.0% (trimethoprim) and 50.0% (tetracycline), respectively, for *E. coli* of RTE meat origin.

### 3.3. Multidrug Resistant of Individual E. coli Isolates

Table 3 shows the multiple antibiotic resistance (MAR) index and antimicrobial resistance pattern of individual *E. coli* species isolated from raw meats, RTE meats and their related samples. The MAR index of the *E. coli* isolates was within the range of 0.0 to 0.7 and displayed 22 resistant patterns. Resistance to A (amoxicillin) and A-T-TM (amoxicillin-tetracycline-trimethoprim) was found in six *E. coli* isolates each and was the most common pattern observed. Resistance to as high as six different classes of antibiotics—that is, A-ATH-T-C-CRO-T and A-T-C-CRO-TM-IMI—was observed for FP9 (*E. coli* isolated from raw pork) and FP6 (*E. coli* isolated from raw pork), respectively. RH7 (*E. coli* isolated from hand swab of raw meat seller), RB4 (*E. coli* isolated from raw beef) and RK7 (*E. coli* isolated from knife swab of raw meat seller) were resistant to five (A-ATH-T-CRO-TM, CIP-A-T-C-CRO and CIP-A-T-C-TM) different classes of antibiotics, respectively. Resistance to zero, one, two, three, four, five and six antimicrobials was 11.1%, 28.9%, 8.9%, 17.8%, 22.2%, 6.7% and 4.4%, respectively. Multidrug resistance (that is, resistant to ≥3 different antimicrobials) was 51.1%.

## 4. Discussions

*Escherichia coli* harbor the GIT of animals including humans and are normally distributed in the environment by these organisms [1,3]. Their distribution among humans, animals and the environment is associated with the re-circulation of antimicrobial resistant and pathogenic strains. Raw meats, ready-to-eat (RTE) meats (except RTE pork), hands of meat sellers, knives used by meat sellers for cutting meat, tables on which meats are sold or cut into pieces and utensils for processing meat were all contaminated with *E. coli* species. This reveals lapses in the processing of the meat and meat products. It also suggests possible cross contamination among the samples examined. It must be noted that knives, tables and utensils are not primary sources of *E. coli* and thus were cross contaminated. *E. coli* has been reported as an indicator organism for revealing unhygienic conditions in the food processing chain [1]. The rate of contamination of raw meats by *E. coli* was higher than the RTE meats, which was expected. The heat treatment RTE meats were subjected to during cooking accounted for this occurrence [36]. The occurrence of *E. coli* in raw meats, RTE meats and their related samples has been reported in other countries. In the USA, Zhao et al. [10] reported that retail chicken (83.5%), beef (68.9%) and pork (44.0%) were contaminated by *E. coli*. Fresh bovine meat (1.0%), fresh ovine (5.3%) and fresh pigs (3.0%) of Europe origin were contaminated with *E. coli* [9]. Again, the EFSA [9] reported that meat preparations and meat products from mixed sources were contaminated by shiga-toxin producing *E. coli* non O157 (2.7%) and shiga-toxin producing *E. coli* (2.2%). Zhao et al. [10] and the EFSA [9] reported lower incidences of *E. coli* as compared to this study. Their works were conducted in countries where adherence to hygienic handling and processing of animals into meat and meat products are given much priority and standards are followed and monitored. The same cannot be said of Ghana. *Escherichia coli* was observed in 32% of RTE meat products collected from Latvia [11], which was within the range (0–50%) found in this study. In Egypt, *E. coli* was detected in raw beef (54%), chicken (16%) and hand swabs from sellers (24%) [16]. According to Parvin et al. [14], 76% of frozen chicken meats collected from Bangladesh were positive for *E. coli*. The results of Gwida et al. [16] in Egypt and that of Parvin et al. [14] in Bangladesh were quite similar to the findings of the current work. These developing countries might have similar situations in the handling of animals and processing of meat and meat products to that of Ghana. However, it is emphasized that factors, such as the accuracy, type of the isolation or detection method employed, number of samples examined, type of samples examined, period or season of sampling, level of hygiene observed or practiced and location of sampling, are responsible for wide differences in the occurrences of *E. coli* and other bacteria in humans, foods, animals and the environment [2,4,9,10]. The presence of *E. coli* in this current study also signifies that meat and meat products are important sources of *E. coli* and possible risk sources for *E. coli* infection.

Thus, the resistance of *E. coli* to antimicrobials in this study should be worrying. This study found high phenotypic resistance of *E. coli* to penicillins (amoxicillin), trimethoprims (trimethoprim), and tetracyclines (tetracycline). The EFSA [42] reported that ampicillin, sulfamethoxazole, tetracycline and trimethoprim were the antimicrobials most often represented in the pattern of multidrug resistance *E. coli* isolates from bovines, porcine and poultry species. Other studies have also reported higher resistances to tetracycline [10,13,14] and ampicillin [12,14]. For example, Zhao et al. [10] reported resistance to tetracycline (50.3%) and gentamicin (18.6%) in *E. coli* isolated from different retail meats in the USA. This study found higher resistance to tetracycline, but lower for gentamicin. In the current study, average phenotypic resistance was higher in the human hands (hands of meat sellers) and working tools (tables, knives and utensils used by meat sellers) *E. coli* isolates than the raw and RTE meats *E. coli* isolates. There were some variations and similarities in the phenotypic resistance patterns among the raw meats, RTE meats, hands of sellers and the working tools used by meat sellers. Phenotypic antimicrobial resistance patterns allow for the identification of emerging or specific patterns of resistance. For instance, the resistance pattern, A-T-TM (amoxicillin-tetracycline-trimethoprim) was shared by *E. coli* isolated from knife swabs from raw meat seller (FK4), table swabs from raw meat seller (FT1), RTE chevon (Rch1) and table swabs from RTE meat seller (RT3) (Table 3). In addition, FT1 and FT8 or Rch1 and Rch5, which are *E. coli* species isolated from table swabs of raw meat sellers or RTE chevon, respectively, exhibited the same resistant pattern of A-T-TM (amoxicillin-tetracycline-trimethoprim). This means that *E. coli* species from similar sources or locations do not necessarily exhibit the same behavior in their resistant capabilities. Although the meat and their related samples were collected at random to avoid biasness and the assignment of a particular result to a vendor, RH4 and RH4-2 were *E. coli* isolated from the hands of an RTE meat seller, but they exhibited different resistant patterns (amoxicillin-gentamicin-trimethoprim versus amoxicillin, respectively). Bacteria in close association are capable of exchanging genes that confer resistance to each other [43]. This can bring about similarities and differences in their resistance patterns. Therefore, it is essential to understand the resistance mechanisms, distribution of resistant determinants in bacteria and to determine humans, animal, food and environmental factors that facilitate their dissemination. Processes such as antibiotic degradation/modification, antibiotic sequestration, antibiotic target modifications and antibiotic target bypass are among the processes proposed as means for transfer and exchange of antimicrobial resistances in bacteria [42,43].

The multiple antibiotic resistance (MAR) index of the *E. coli* isolates ranged from resistant to zero antibiotics to resistant to six antibiotics. It has been shown that bacteria with MAR index of >0.2 originate from high-risk sources, that is, sources where growth promoters or several antibiotics are used, whilst MAR index of <0.2 are isolates originating from sources where less antibiotics are used [44]. In the present study, 50.1% had MAR index between 0.3 to 0.7, meaning that most *E. coli* isolates were isolated from sources exposed to antibiotics and their usage. A much higher percentage of the *E. coli* exhibited multidrug resistance. Two, three, ten and eight *E. coli* isolates were resistant to six, five, four and three different classes of antimicrobials, respectively. Multidrug resistance *E. coli* isolates from raw meats, RTE meats and/or their environment have also been reported by Zhao et al. [10], Parvin et al. [14], EFSA [9] and Altalhi et al. [12] in different countries. Zhao et al. [10] found that *E. coli* isolates from chicken (38.9%), pork (17.3%) and beef (9.3%) were multidrug resistant. Parvin et al. [14] reported that all *E. coli* isolated from frozen chicken meats were multidrug resistant. In addition, Altalhi et al. [12] indicated that 15 strains of *E. coli* isolated from retail raw chicken meat exhibited multidrug resistance phenotype and harbored at least three resistance genes. Resistance genes, such as bla_CTX-M_ and bla_TEM_ (beta-lactamase), *tet (A)* and *tet (B)* (tetracycline), *aph (3’’)-Ib* and *apha (6)* (aminoglycoside), *dfrA17* and *dfrA14* (trimethoprim) and *mdf(A)* macrolide, have been found in *E. coli* isolated from Ghana [23,24]. In addition, virulence factors, such as *lpfA, gad, mchF, iha, mchB, mchC eilA, iss, cftB* and *mcmA*, have been associated with *E. coli* isolates from various meat samples in Ghana [45], revealing the potential and possibility of *E. coli* isolates from this study to be pathogenic, although this study did not seek to establish that. The developments of resistances are elicited by factors including the inappropriate use of antimicrobials and poor hygiene practices in healthcare and food chain setups [19,20], facilitating the transfer of resistant *E. coli*. *E. coli* resistant to antimicrobials present in meats and meat products can spread to humans via consumption. They can also infest humans via contamination from their own hands and working tools. Multiple antimicrobial-resistant *E. coli* will make antimicrobials less effective against them and result in treatment failures [9,19]. *E. coli* is a commensal organism and can also serve as a reservoir for antimicrobial-resistant genes among other commensals and pathogenic bacteria [9,20]. Monitoring antimicrobial resistance in *E. coli* and other bacteria in meats, meat products and foods is required to elucidate the reason for which they develop and spread resistance among humans, animals, the environment and food chain. 

## 5. Conclusions

The raw meats, RTE meats, and hands, tables, knives and utensils of meat sellers were contaminated by *E. coli*, revealing possible unhygienic practices. This poses the risk of *E. coli* infection. Raw beef and mutton were the most contaminated source. *E. coli* was not detected in RTE pork. Generally, raw meats were more contaminated with *E. coli* than the RTE meats, warranting appropriate cooking or heating prior to consumption. Most (50.1%) of the *E. coli* exhibited resistance to multiple antimicrobials. This is a sign of emerging resistances and reduction in the effectiveness of antimicrobials. Resistance was high for amoxicillin, tetracycline and trimethoprim. Resistance to only amoxicillin or amoxicillin-tetracycline-trimethoprim was most common. Hygienic practices among fresh and RTE meat sellers is recommended to prevent the spread of antimicrobial-resistant *E. coli* in the food chain and among humans, animals and their environment. Molecular characterization and detection of ESBL genes of the *E. coli* isolates is recommended.

## Figures and Tables

**Figure 1 microorganisms-09-00326-f001:**
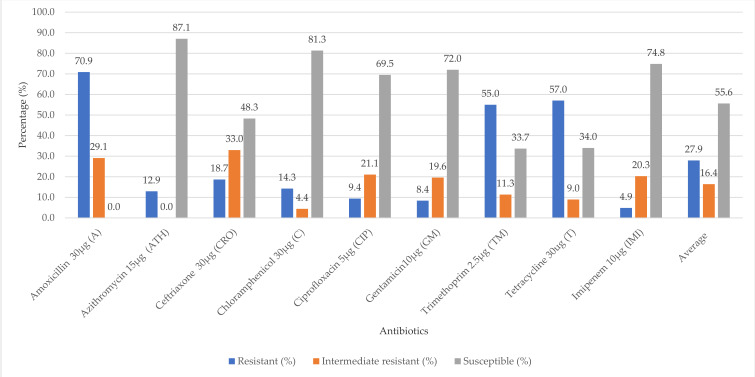
Overall antibiotic susceptibility of *E. coli* (*n* = 45) isolated from raw meats, RTE meats and their related samples.

**Table 1 microorganisms-09-00326-t001:** Distribution of *Escherichia coli* in raw meats, ready-to-eat (RTE) meats and their related samples.

Source	No. Tested	No. Positive	Prevalence (%)
Raw beef	10	8	80
Raw chevon	10	6	60
Raw chicken	10	3	30
Raw guinea fowl	10	4	40
Raw mutton	10	7	70
Raw pork	10	6	60
RTE beef	10	5	50
RTE chevon	10	5	50
RTE chicken	10	2	20
RTE guinea fowl	10	2	20
RTE mutton	10	2	20
RTE pork	10	0	0
Hand swab of raw meat sellers	10	1	10
Knife swab from raw meat sellers	10	6	60
Table swab from raw meat sellers	10	3	30
Utensil swab from raw meat sellers	10	1	10
Hand swab from RTE meat sellers	10	5	50
Knife swab from RTE meat sellers	10	3	30
Table swab from RTE meat sellers	10	3	30
Utensil swab from RTE meat sellers	10	3	30
Overall	200	75	38

**Table 2 microorganisms-09-00326-t002:** Phenotypic antibiotic resistance of *E. coli* (*n* = 45) isolated raw meats, RTE meats, hands and working tools of meat sellers.

Antimicrobial	Raw Meat Samples (*n* = 17)			RTE Meat Samples (*n* = 10)			Human Hand Samples (*n* = 5)			Working Tools (*n* = 13)		
	R (%)	I (%)	S (%)	R (%)	I (%)	S (%)	R (%)	I (%)	S (%)	R (%)	I (%)	S (%)
Amoxicillin 30 μg (A)	58.8	41.2	0.0	60.0	40.0	0.0	80.0	20.0	0.0	84.6	15.4	0.0
Azithromycin 15 µg (ATH)	11.8	0.0	88.2	20.0	0.0	80.0	20.0	0.0	80.0	0.0	0.0	100.0
Ceftriaxone 30 µg (CRO)	29.4	23.5	47.1	10.0	50.0	40.0	20.0	20.0	60.0	15.4	38.5	46.2
Chloramphenicol 30 µg (C)	29.4	17.6	52.9	20.0	0.0	80.0	0.0	0.0	100.0	7.7	0.0	92.3
Ciprofloxacin 5 µg (CIP)	0.0	23.5	76.5	10.0	10.0	80.0	20.0	20.0	60.0	7.7	30.8	61.5
Gentamicin 10 µg (GM)	5.9	0.0	94.1	0.0	40.0	60.0	20.0	0.0	80.0	7.7	38.5	53.8
Trimethoprim 2.5 µg (’TM)	52.9	17.6	29.4	30.0	0.0	70.0	60.0	20.0	20.0	76.9	7.7	15.4
Tetracycline 30 µg (T)	41.2	5.9	52.9	50.0	10.0	40.0	60.0	20.0	20.0	76.9	0.0	23.1
Imipenem 10 µg (IMI)	11.8	23.5	64.7	0.0	30.0	70.0	0.0	20.0	80.0	7.7	7.7	84.6
Average	26.8	17.0	56.2	22.2	20.0	57.8	31.1	13.3	55.6	31.6	15.4	53.0

RTE, Ready-to-eat. S, Susceptible; I, Intermediate resistance; R, Resistant. Raw meat: raw beef, chevon, chicken, guinea fowl, mutton and pork. Ready-to-eat meat: raw beef, chevon, chicken, guinea fowl and mutton. Human hand samples: hand swab from raw and ready-to-eat meat sellers. Working tools: knives, tables and utensils of raw and ready-to-eat meat sellers.

**Table 3 microorganisms-09-00326-t003:** Multiple antibiotic resistance index and antimicrobial resistance profile of individual *E. coli* species of raw meats, ready-to-eat (RTE) meats, and their related samples.

MAR Index	Antibiotic Resistant Profile	Source	Code
0.7	A-ATH-T-C-CRO-T	Raw pork	FP9
0.7	A-T-C-CRO-TM-IMI	Raw pork	FP6
0.6	A-ATH-T-CRO-TM	Hand swab of raw meat seller	RH7
0.6	CIP-A-T-C-CRO	RTE beef	RB4
0.6	CIP-A-T-C-TM	Knife swab from raw meat seller	FK7
0.4	A-T-CRO-TM	Raw beef	FB8
0.4	A-T-CRO-TM	Utensil swab from RTE meat seller	RU9
0.4	A-T-C-TM	Raw beef	FB1
0.4	A-T-C-TM	Raw mutton	FM4
0.4	A-T-C-TM	RTE beef	RB1
0.4	A-T-GM-TM	Raw beef	FB4
0.4	A-T-GM-TM	Knife swab from raw meat seller	FK9
0.4	A-T-TM-IMI	Raw chevon	Fch7
0.4	A-T-TM-IMI	Table swab from raw meat seller	FT9
0.4	CIP-A-T-TM	Hand swab of raw meat seller	FH2
0.3	A-CRO-TM	Raw chicken	FC1
0.3	A-GM-TM	Hand swab from RTE meat seller	RH4
0.3	A-T-TM	Knife swab from raw meat seller	FK4
0.3	A-T-TM	Table swab from raw meat seller	FT8
0.3	A-T-TM	RTE chevon	Rch1
0.3	A-T-TM	RTE chevon	Rch5
0.3	A-T-TM	Table swab from RTE meat seller	RT3
0.3	A-T-TM	Table swab from RTE meat seller	RT4
0.2	A-T	RTE mutton	RM5
0.2	A-TM	Table swab from raw meat seller	FT1
0.2	CRO-TM	Raw guinea fowl	FG5
0.2	T-CRO	Knife swab from RTE meat seller	RK6
0.1	A	Raw chicken	FC4
0.1	A	Raw chevon	Fch5
0.1	A	Table swab from raw meat seller	FT3
0.1	A	Utensil swab from raw meat seller	FU10
0.1	A	RTE beef	RB8
0.1	A	Hand swab from RTE meat seller	RH4-2
0.1	ATH	Raw guinea fowl	FG3
0.1	ATH	RTE chicken	RC9
0.1	ATH	RTE chevon	Rch9
0.1	C	Raw mutton	FM5
0.1	T	Hand swab of raw meat seller	FH1
0.1	TM	Raw chevon	Fch2
0.1	TM	Knife swab from raw meat seller	FK1
0.0	None	Raw mutton	FM1
0.0	None	Raw mutton	FM9
0.0	None	Raw pork	FP1
0.0	None	RTE chicken	RC6
0.0	None	RTE guinea fowl	RG9

A, Amoxicillin 30μg; ATH, Azithromycin 15µg; CRO, Ceftriaxone 30µg; CIP, Ciprofloxacin 5µg; GM, Gentamicin 10µg, TEC, TM, Trimethoprim 2.5µg; T, Tetracycline 30ug, IMI, Imipenem 10µg.

## Data Availability

No new data were created or analyzed in this study.

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
