# Peer review of "Phenotypic Antimicrobial Susceptibility of Escherichia coli from Raw Meats, Ready-to-Eat Meats, and Their Related Samples in One Health Context"

_microorganisms, 2021, doi:10.3390/microorganisms9020326_

Round 1

Reviewer 1 Report

The work in its current form presents itself more scientifically than during the first submission to the journal and discusses important issues.
I am still concerned about the identification of bacteria and confirmation of the isolated bacteria with the PCR test. It would be necessary to extend this type of analysis to include accurate identification and characterization of bacteria. I also understand that the experimental setup did not assume this, and I leave it to the editor's decision to publish the article.
Completed content I consider satisfactory.

Author Response

Thank you. We are 100% sure our E. coli isolates were scientifically and correctly identified by the conventional method we employed.

Reviewer 2 Report

Dear Authors,

I feel that my previous concerns have been adequately addressed and the manuscript have improved considerably.

However English language use still need to be improved and there are still some issues that needs to be addressed:

  • L74 Please change to: Also, farmers did not demonstrate in-depth knowledge in the antimicrobials used
  • L75-76 Please rephrase
  • L80 Please correct "haboured" to "harboured" and specify the resistance genes found
  • L83 Again please be more specific
  • L204 Definition of multidrug resistance should be placed in the Materials and Methods section and reference should be given (Magiorakos et al., 2012)
  • L244-246 Please rephrase this sentence. The presence of E. coli contamination in meat and meat products do not necessarily result in infections.
  • L263 The codes used can't be understood since not present anymore in the current version of the manuscript. I suggest to simply refer to samples obtained from RTE chevon and table swabs.
  • L268-269 As above
  • L270 Please rephrase and provide reference.
  • L275-278 Antibiotic efflux, antibiotic degradation/modification, antibiotic sequestration, antibiotic target protection, antibiotic target modifications and antibiotic target bypass are resistance mechanisms to antimicrobials. Please rephrase.
  • L290 there is no resistance to multidrugs. Please change to multidrug resistant.
  • L303-304 Please check this sentence.
  • L305-308 Please consider to rephrase.

Author Response

I feel that my previous concerns have been adequately addressed and the manuscript have improved considerably. Thank you.

However English language use still need to be improved and there are still some issues that needs to be addressed: The English has been cross-checked and corrected appropriately.

  • L74 Please change to: Also, farmers did not demonstrate in-depth knowledge in the antimicrobials used. Done, see line 74.
  • L75-76 Please rephrase. Done, see line 75.
  • L80 Please correct "haboured" to "harboured" and specify the resistance genes found, Done, see lines 80 and 81.
  • L83 Again please be more specific. Done, see lines 83 and 84.
  • L204 Definition of multidrug resistance should be placed in the Materials and Methods section and reference should be given (Magiorakos et al., 2012). Done see line 146.
  • L244-246 Please rephrase this sentence. The presence of  colicontamination in meat and meat products do not necessarily result in infections. Done see line 245 to 247.
  • L263 The codes used can't be understood since not present anymore in the current version of the manuscript. I suggest to simply refer to samples obtained from RTE chevon and table swabs. Thank you. Actually the codes are present at the last column of Table 3. The codes help to differentiate E. coli from the same source. See lines 263 and 264.
  • L268-269 As above. We actually left the codes at the last column of Table 3 for differentiation of E. coli isolates from the same source. See lines 264 and 270.
  • L270 Please rephrase and provide reference. Done see line 272 and 273.
  • L275-278 Antibiotic efflux, antibiotic degradation/modification, antibiotic sequestration, antibiotic target protection, antibiotic target modifications and antibiotic target bypass are resistance mechanisms to antimicrobials. Please rephrase. Done see, lines 276 to 278.
  • L290 there is no resistance to multidrugs. Please change to multidrug resistant. Done, see line 290.
  • L303-304 Please check this sentence. Done, see lines 306 to 307.
  • L305-308 Please consider to rephrase. Done, see lines 306 to 309.

This manuscript is a resubmission of an earlier submission. The following is a list of the peer review reports and author responses from that submission.

Round 1

Reviewer 1 Report

In the manuscript titled: “Phenotypic antimicrobial susceptibility of Escherichia coli from raw meat, ready-to-eat meat, and their related samples in one health context”, the authors assess the presence of E. coli in various products and surfaces related to the handling to meat products and further assess the antimicrobial resistance profiles of the 75 E. coli isolates that were detected from 200 samples.

The authors provide a descriptive study to assess the risk of E. coli as an indicator for the potential of food-borne illness and drug resistant infections and/or spread in Northeast Ghana. Antimicrobials susceptibilities were compared to determine the frequency of multidrug resistance phenotypes.

Major concerns:

While this is a descriptive study, the discussion of the data observed is overall lacking, and it greatly improve the quality and significance of this article to expand on the discussion of the data:

  • Lines 170-177 the authors discuss the presence of coli in various meats from different regions (such as US vs. Europe) but provide the reader with no analysis of why the massive variation is detected worldwide, and where the study fits within this potential expanse of results. More discussion on this point is warranted.

  • Lines 196-197   Similar sources do not necessarily share resistance profiles, however one factor not addressed in this discussion is if samples taken from the same vendor or marketplace (though different locations, such as raw meat and the knife and hands of the person selling that meat) had similar profiles. This would be a valuable point to make in this paper as a whole. And would strengthen the analysis.

  • If the authors would like to infer the increased risk of transmission of the antimicrobial resistance traits, that would be best tested by determining the resistance mechanism – though I understand this is beyond the scope of this paper. However, some examples of resistance mechanisms that could account for the resistance profiles observed would enhance the discussion on this point.

Minor concerns:

Line 86 – perhaps “well” should be “were”

Line 129: The E. coli isolates were The Escherichia coli isolates were

Line 138 - it is unclear what ≥ 55 indicates

Line 145 – define MAR index to the reader. What it is and it’s significance – perhaps this should be either described here in the discussion or previously in the introduction

Line 152 – it can be inferred the multidrug resistance here was defined as resistance to ≥ 3 antimicrobials (as is customary), but it would be beneficial to state that here for the ease of the reader.

Tables & Figures:

Table 1 – drop the decimal value for %, there is not enough power to make the decimal value significant.

Figure 1 – Clearly state that n = 75 (I believe) E. coli isolates

  • Rather than alphabetical – suggest to order drugs by drug class
  • Some numbers do not make sense – if n = 75

Table 2 – Strongly suggest to include the n or each group: e.g.: Raw meat (n = 34)

Table3 – Suggest to reorder the rows by MAR value then grouping by antimicrobial resistance profile, resistances or source?

Reviewer 2 Report

The publication presents a very important issue. Nevertheless, the analysis is very simple and does not present a comprehensive solution to the problem that we should strive for in the current research. There are also some gaps in the text.
The introduction does not introduce the reader to the issue related to E. coli. There is no specific description of the problem - in its present form it has a popular science character. Describe the exact bacteria in the context of commensal and harmful strains. What is the importance of E. coli, what are the general characteristics of E. coli.

Why was the isolated bacteria not confirmed by PCR tests? Identification of the bacteria is also missing. It would be necessary to extend this type of analysis to include accurate identification and characterization of bacteria.

The discussion is about presenting the results, not a fair comparison with the literature.

Reviewer 3 Report

Dear Authors,

The manuscript presents some interesting results regarding E. coli rate of contamination of raw meat, ready-to-eat meat and other high risk surfaces the Upper East Region of Ghana. The manuscript is well written and clearly presents research outputs of contamination rate, antimicrobial resistance frequency and the resistance phenotype of isolates. The study design is adequate for this type of investigation and statistical analysis is appropriate. 

Please consider to revise the introduction section and introduce the relevance of AMR in foodborne pathogens to readers. This has been better presented in the discussion sections. One health is mentioned only in the title of the manuscript without further mention in the text. I completely agree that AMR in foodborne pathogens is of great importance for One Health, but this deserve a better introduction. 

My major concerns are related to the antimicrobial susceptibility testing of the isolates:

  • The panel used do not allows the identification of ESBL producing bacteria, therefore antimicrobial susceptibility results might be inaccurate.
  • Why teicoplanin was included? It's an antibiotic with activity mainly towards Gram positive bacteria and E. coli are intrinsically resistant to it. 
  • Please check breakpoint reference and update it if possible. For animals isolates the current reference is: VET01: Performance Standards for Antimicrobial Disk and Dilution Susceptibility Tests for Bacteria Isolated From Animals, 5th Edition.
  • Lack of molecular characterization is a major limitation of the study since limited informations can be drawn regarding the genetic background of the isolates.
  • No explanation of the MAR index present in Table 3 is given in the text. Please include reference and details in the Materials and methods section. 

Please consider to include more relevant literature in the discussion. For example:

  • Dsani, E., Afari, E.A., Danso-Appiah, A. et al. Antimicrobial resistance and molecular detection of extended spectrum β-lactamase producing Escherichia coli isolates from raw meat in Greater Accra region, Ghana. BMC Microbiol 20, 253 (2020)
  • Eibach, Daniel, et al. "Extended-spectrum beta-lactamase-producing Escherichia coli and Klebsiella pneumoniae in local and imported poultry meat in Ghana." Veterinary microbiology 217 (2018): 7-12.